# Spurious Features in Continual Learning

Timothée Lesort

UdeM, Mila AI Institute.

## Abstract

Continual Learning (CL) is the research field addressing training settings where the data distribution is not static. One of the core problems CL addresses is learning without forgetting. To solve problems, continual learning algorithms need to learn robust and stable representations based only on a subset of the data. Those representations are necessarily biased and should be revisited when new data becomes available. This paper studies spurious features' influence on continual learning algorithms. We show that in continual learning, algorithms have to deal with local spurious features that correlate well with labels within a task only but which are not good representations for the concept to learn. One of the big challenges of continual learning algorithms is to discover causal relationships between features and labels under distribution shifts.

## Introduction

Feature selection is a standard machine learning problem. Its objective is notably to improve the prediction performance (Guyon and Elisseeff 2003). In the presence of spurious features, a learning algorithm may overfit features and learn a solution that can not generalize to the test set. This problem can notably be caused by a covariate shift between train and test data.

In continual learning (CL) (French 1999; Parisi et al. 2019; Lesort et al. 2020), the training data distribution changes through time. Hence, spurious features (SFs) in one time-step of the data distribution should not last. A CL algorithm relying on a spurious feature could then be resilient and learn better features later – given more data. Algorithms can also learn to ignore past spurious features (Javed, White, and Bengio 2020). An example of a task with spurious features could be a classification task between cars and bikes. In the training data, all cars are red, and all bikes are white, while it test data, both are in a unique blue not available in train data. A model could easily overfit the color to solve the task while it is not discriminative in the test data. Addressing spurious features was one of the major goals of the recent out-of-distribution (OOD) generalization community (Arjovsky et al. 2019; Ahuja et al. 2021; Sagawa et al. 2019; Pezeshki et al. 2020).

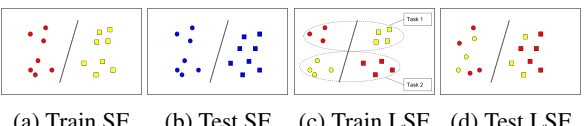

(a) Train SF    (b) Test SF    (c) Train LSF    (d) Test LSF

Figure 1: **Spurious features and local spurious features.** If the task is to distinguish the squares from the circles. In Fig. 1a and 1b, the color is a spurious feature because there is a covariate shift between train and test data. In Fig. 1c and 1d, we observe two tasks of a domain-incremental scenario, the colors are locally spurious in tasks 1 and 2. Even if there is no significant covariate shift between train and test full data distribution, colors appear discriminative while looking at data within a task.

On the other hand, in continual learning, the second type of spurious feature can be described: *local spurious features*. Local features denote features that correlate well with labels within a task (a state of the data distribution) but not in the full scenario. In opposite to the usual *spurious features*, this problem is provoked by the unavailability of all data. An example of a classification scenario would be: task 1, blue cars vs yellow bikes and task 2, yellow cars vs blue bikes. In both cases, the tasks can be solved efficiently with the colour feature, but if the test data is composed of cars and bikes of both yellow and blue, then the colour is not discriminative anymore, and the model can not generalize. While there is no covariate shift between all train data and test data, the model can not generalize because of the distribution shift through time. It is, therefore, a problem specific to continual learning.

This paper investigates the problem of spurious features (with covariate shift) in continual learning and shows that the continual learning setup leads to a specific type of spurious features that we call local spurious features (LSFs) (without covariate shift) in CL as shown in Fig. 1.

## Problem Formulation

This section introduces the spurious features problems in a sequence of tasks. The goal is to present the key types of features, namely: general, local, and spurious features.

**General Formalism:** We consider a continual scenario of classification tasks. We study a function $f_\theta(\cdot)$, imple-

Table 1: Summary of characteristics of the types of features. For a feature $z$ of a class $c$, we denote if it verify (1) on different data setting, a single task $\mathcal{T}_t$, the whole scenario $\mathcal{C}_T$, the test set $\mathcal{D}_{te}$.

| Name | $\mathcal{T}_t$ | $\mathcal{C}_T$ | $\mathcal{D}_{te}$ |
|---|---|---|---|
| Good Feature ($z_+$) | ✓ | ✓ | ✓ |
| Spurious Feature ($z_{spur}$) | ✓ | ✓ | ✗ |
| Local Feature ($z_{loc}$) | ✓ | ? | ? |
| Local Spurious Feature ($z_{spur:t}$) | ✓ | ✗ | ✗ |

mented as a neural network, parameterized by a vector of parameters $\theta \in \mathbb{R}^p$ (where p is is the number of parameters) representing the set of weight matrices and bias vectors of a deep network. In continual learning, the goal is to find a solution $\theta^*$ by minimizing a loss $L$ on a stream of data formalized as a sequence of tasks $[\mathcal{T}_0, \mathcal{T}_1, ..., \mathcal{T}_{T-1}]$, such that $\forall (x_t, y_t) \sim \mathcal{T}_t$ ($t \in [0, T-1]$), $f_{\theta^*}(x) = y$. We do not use the task index for inferences (i.e. single head setting).

To describe the different types of features, let $z$ be a feature and $x \sim \mathcal{D}$ a datum point in dataset $\mathcal{D}$. We define $w(.)$ a function which returns 1 if $z$ is in $x$ and 0 if not. $w(.)$'s output is binary for simplicity. Then, for all data with a label $y$ in the dataset $\mathcal{D}$, we can compute the correlation $c(\mathcal{D}, z, y) = correlation(w(z, x) = 1, Y = y)$, which estimates how a feature correlates with the data of a given class. We can then define discriminative features as:

$z$ is discriminative for class $y$ in $\mathcal{D}$ if:

$$\forall y' \in \mathcal{Y}, y \neq y' \quad c(\mathcal{D}, z, y) \gg c(\mathcal{D}, z, y') \qquad (1)$$

$\mathcal{Y}$ is the set of classes in $\mathcal{D}$. In other words, $z$ is discriminative for $y$ if it correlates significantly more to $y$'s data than to the data of any other class. Then a good feature $z_+$ for a class $y$ respects (1) for training data $\mathcal{D}_{tr}$ and test data $\mathcal{D}_{te}$.

**Spurious Features vs Local Spurious Features**

A spurious feature $z_{spur}$ for a class $y$ respects (1) for training data $\mathcal{D}_{tr}$ but not for test data $\mathcal{D}_{te}$. A spurious feature is well correlated with labels in training data but not with testing data.

Hence, learning from $z_{spur}$ may offer a low training error but high test error. The presence of $z_{spur}$ is due to a covariate shift between train and test distribution which changes the feature distribution.

In continual learning, the covariate shift between train and test $z_{spur}$ may also lead to poor generalization. Further, the features can be locally spurious, e.g., they correlate well with labels within a task but not within the whole scenario. We name them *local spurious features* (LSF). We illustrate the difference between spurious features and local spurious features in Figure 1.

At task $t$, A local spurious feature $z_{spur;t}$ respects (1) for a class $y_t$ in task $\mathcal{T}_t$, but not for the whole scenario $\mathcal{C}_T$. $z$ is a LSF for a class $y$ in $\mathcal{T}_t \sim \mathcal{C}_T$, with $t \in [\![0, T-1]\!]$:

$$\text{if } \forall \, y' \in \mathcal{Y}_t, y \neq y' \quad c(\mathcal{T}_t, z, y) \gg c(\mathcal{T}_t, z, y')$$
$$\text{and } \exists \, y'' \in \mathcal{Y}, y \neq y'' \quad c(\mathcal{C}_T, z, y) \not\gg c(\mathcal{C}_T, z, y'') \qquad (2)$$

$\mathcal{Y}_t$ is the classes set in task $\mathcal{T}_t$ and $\mathcal{Y}$ is the classes set in the full scenario $\mathcal{C}_T$ composed of $T$ tasks. A LSF $z_{spur;t}$

correlates well with a label on the current task but not on the whole scenario. $z_{spur;t}$ can be extended from a single task $\mathcal{T}_t$ to all task seen so far $\mathcal{T}_{0:t}$ without loss of generality.

**Global vs Local Optimum:** We assume that machine learning models solve tasks by learning to detect/select features that correlate well with labels. Then, while learning on a task $t$, we distinguish a local optimum $\theta_t^*$, satisfying for the current task $\mathcal{T}_t$, from a global optimum $\theta_{0:T}^*$ that is satisfying for whole scenario $\mathcal{C}_T$ (past, current, and future tasks).

Similarly, we can differentiate local and global features, leading to local and global optimum. The global features are the good features $z_+$ that are predictive for the full scenario. Unfortunately, at time $t$, we can not know if a feature is part of $z_+$ without access to the future data. Therefore, algorithms should learn with their current data but update their knowledge afterwards, given new data. For example, in classification, the discriminative features for a given class depend on all the classes. Therefore, when new classes arrive, discriminative features can become outdated in class-incremental scenarios.

To learn robust solution in CL, algorithms should them be able to deal both with spurious features and local spurious features. One trivial solution to deal with local spurious features is the use of replay. Replay can avoid and fix local spurious features' influence by providing more context on the full data distribution. Nevertheless, replay can be compute and data-intensive and a better solution could be developed.

## Conclusion

Continual learning algorithms are built to learn, accumulate and memorize knowledge through time to reuse them later. Memorizing bad features can have catastrophic repercussions on future performance. Then, to learn general features, algorithms need to deal with spurious and local spurious features.

This paper first investigates the question of spurious features on continual learning. Algorithms easily overfit spurious features for one or several tasks, leading to poor generalization. Spurious features are then problematic for them. Furthermore, we formalize another type of spurious feature that we call local spurious feature and which can be problematic for continual learning algorithms.

Local spurious features are features that correlate well with labels when only a subset of data are available but not when all the data is available. These types of features make harder the discovery of robust features. From a causality perspective, local spurious features makes it harder to discover the causal relationship between features and labels in continual learning. Causality algorithms could help to find a solution to solve this issue.

In the continual learning literature, performance decrease is generally attributed to catastrophic forgetting. Our results show that the problem of local spurious features also plays a major role. More research is needed to understand better the impact of local spurious features along with catastrophic forgetting. Understanding this phenomenon is critical to better address forgetting and feature selection and enable efficient continual learning.

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
