# OpenReview forum: "Spurious Features in Continual Learning"
_AAAI.org/2023/Bridge/CCBridge — AAAI23 Bridge Continual Causality_

### Official Review · Reviewer_UnVk · 2022-12-02
**Interesting direction but the link to causality should be made stronger**

**Rating:** 7
**Confidence:** 3

**Review:**

The paper considers spurious features' influence on continual learning algorithms. A bit like confounders well-known in classifications tasks, it shows that there might be features that correlate well with labels within one task only but which are not good representations for the concept to learn within continual learning. These local spurious features are specific to continual learning, and have not ben considered in the literature so far. Overall, I like the direction for a workshop like program such as the bridge on continual and causal learning. However, making the link to causality stronger might be good. For instance, is it fair to treat the local spurious features as a kind of confounder? Probably not, but I am just wondering. Also, I am wondering about potential connections to continual learning and novelty resp. OOD detection. There is also some work on adversarial attacks on continual learning. Is this related? Isn't this also indicating that there are local spurious features? Anyhow, I like the direction but the connection to causality should be improved.

---

### Official Review · Reviewer_TJrT · 2022-12-03
**Recommending acceptance since it is an important topic for the program**

**Rating:** 7
**Confidence:** 4

**Review:**

My reviews are based on these instructions sent in the email:
> Their [reviews] primary purpose is to check for factual correctness and broad relation to the bridge topics. As a guideline, we envision the reviews to be inclusive and include suggestions on the laid-out directions, rather than voicing direct critique at such an early stage.

**Relevance for the program: (High)**

Dealing with spurious (local/global) features in the continual learning setting is a highly relevant topic for this program. This setup reflects the challenge of the real world where the data stream may contain global/local spurious features. This setup may inspire the design of robust causal models. At the very least, this is a logical evaluation setup to evaluate the generalization of such models.

**Factual Correctness: (Could be improved/clarified better)**

The premise is factually correct i.e., it is reasonable to assume that data streams will contain global/local spurious features.

However, the relation to causal models and the exact position this paper takes is not well-clarified. Apart from a few sentences in the abstract/conclusion, I did not find a discussion on the role of causal models. Some topics that might be worth discussing: would causal models require different strategies to deal with global vs local spurious cues? can the transient nature of local features itself help identify core features? etc

Also, it would be more helpful to discuss spurious features + continual learning + causal modeling using realistic examples e.g., medical diagnosis with devices whose noise characteristics may change over time; object recognition for autonomous vehicles under unseen future weather patterns, contexts etc.

Having said that, the setting itself is highly relevant to the program and might inspire discussions on building/evaluating robust causal models.

---

### Official Review · Reviewer_tddS · 2022-12-03
**Important Discussion on Spuriousness in (Continual) Learning Settings**

**Rating:** 6
**Confidence:** 3

**Review:**

The discussion in this short paper is as crucial for continual causality as it is for machine learning per se.

Spurious features are a real problem for learning systems that are required to reason in the face of new data distributions influence by, say, covariate shift. Why? Because spurious features are features that act as a "shortcut" to gaining performance on a given task, they are not the real *cause* of the prediction of interest. Here causality comes into play. Spurious features or rather spurious association as referred to typically within the (Pearlian) causality literature is about situations in which confounding (a common cause to what is being used for predicting and what is being predicted) occurs and messes with our conclusions through pathways in the causal graph that allow for information flow aside from the actual cause-effect relationship of interest.

The presented short paper discusses a *new type* of spurious feature, referred to as *local* spurious feature (LSF), which occurs as an important artefact of looking at continual learning, more specifically the setup of sequential task solving. The authors provide a high-level idea in Figure 1, where they argue that within tasks on the training data there could be spurious features that are not present across train and test data. To rephrase, they suggest that the appearance of spurious features is a consequence of the scope of the learning setup, and since in this particular continual learning setup we solve tasks sequentially, the tasks might separately allows for such spurious features. To support their idea, the authors provide some initial formalism using a measure of "discriminativeness" for a given feature w.r.t. some data set and some target (the authors consider general regression of the form f(x)=y). The table on page 2 provides a nice high level overview of "good" features, regular, local, spurious ones and how they relate to train, test and sub-task data.

"Factual correctness:" No critical flaws were found, neither in the overall textual description, nor in the presented formalism. The formalism is not exact (e.g. >> not being defined, C_T not being formalized, c(...)=correlation(...) inaccurate).

"Understandable:" Very much so. IMHO a great addition for a camera-ready version would be a walkthrough on a specific, but relevant application i.e., taking, say, computer vision and showing how certain tasks might look like and what the spurious features across tasks but not across data sets could be.

"Bridging CL and Causality": Weak. Spuriousness, without a doubt, is considered a question of causality, but this is where the "causal" part ends. That is, the presented paper does not make use of any of the existing (Pearlian) causal modelling techniques to discuss the CL setup.

Points of improvement IMHO worth thinking about: (i) an example walkthrough to highlight concretely the relevance of the problem as compliment to the existing abstract formulation, (ii) use Pearl's Causal Hierarchy (L1-3) to discuss how the different settings of traing,test,task can lead to LSF, (iii) provide a motivation for how the research on LSF should be continued (as of now, it seems that the 2 page proposal would be the end of the road in the sense that LSF are simply defined on tasks within the training data but it is not clear what possibilities are available on resolving the LSF caused issues).

---

### Official Review · Reviewer_MoFD · 2022-12-05
**Good idea, but there are some fundamental aspects to improve**

**Rating:** 6
**Confidence:** 5

**Review:**

In this short paper, the authors claim that continual learning problems can be affected by the presence of a specific type of "local spurious feature". An example is given in Fig. 1 which makes intuitive sense. The formal definition is based on Eq. 1 where a feature z is discriminative when very "high" correlation to the feature is observed for the correct class compared to other classes. Then this definition is used in Eq. 2 to define "local" spurious feature.

I think all of this is okay, but this definition does not seem conclusive since it's based on one single feature. In reality, multiple features can trigger correlations and the definition does not include that, and it is not clear how to extend it to multiple features. This is perhaps something to improve in the future.

I also will recommend to improve the abstract. The first line is not correct: CL is not a field to address "forgetting when the data distribution is not static". There are more general definition that can be used. So please rewrite this for the future. Also the last line is unclear. I am not sure why "causal relationships" between features and labels is necessary. And I don't understand why this sentence suddenly appears. The rest of the paper talks very little about causality and simply using correlation. So this seems unnecessary at this point.

---

### Decision · Program_Chairs · 2022-12-05

**Decision:**

Accept

**Comment:**

Accept - Poster

The paper makes an interesting discussion on spuriousness in continual learning settings and presents a new type of spurious feature called local spurious feature (LSF). This topic is of extreme importance in both causality and continual learning settings, making this paper a good fit for the bridge. However, the current presentation of the paper does not include a clear and strong connection with the causal inference literature. We strongly suggest the authors read the reviews carefully and incorporate the link with causality in the camera-ready version of the paper, possibly using concrete examples (see, for example, the points of improvement suggested by Reviewer tddS).  Moreover, consider the reviewer’s suggestions on how to improve the rigor, correctness, and clarity of some statements in the paper. See, for example, the suggestions by Reviewer MoFD regarding the abstract.